# Femtosecond Laser Fabricated Apodized Fiber Bragg Gratings Based on Energy Regulation



**Qi Guo** [1], **Zhongming Zheng** [2], **Bo Wang** [1], **Xuepeng Pan** [1], **Shanren Liu** [1], **Zhennan Tian** [1], **Chao Chen** [2] **and Yongsen Yu** [1,*]

1  State Key Laboratory of Integrated Optoelectronics, College of Electronic Science and Engineering, Jilin University, 2699 Qianjin Street, Changchun 130012, China; qiguo18@mails.jlu.edu.cn (Q.G.); wangbo20@mails.jlu.edu.cn (B.W.); panxp20@mails.jlu.edu.cn (X.P.); liusr19@mails.jlu.edu.cn (S.L.); zhennan_tian@jlu.edu.cn (Z.T.)
2  Changchun Institute of Optics, Fine Mechanics and Physics, Chinese Academy of Sciences, Changchun 130033, China; zhengzm@ciomp.ac.cn (Z.Z.); chenc@ciomp.ac.cn (C.C.)
*  Correspondence: yuys@jlu.edu.cn

**Abstract:** In this paper, an energy regulation method based on the combination of a half-wave plate (HWP) and a polarization beam splitter (PBS) is proposed for the fabrication of apodized fiber gratings, which can effectively improve the side lobe suppression ratio of high-reflectivity fiber Bragg gratings (FBGs) fabricated by femtosecond laser. The apodized FBGs prepared by this method has good repeatability and flexibility. By inputting different types of apodization functions through the program, the rotation speed of the stepping motor can be adjusted synchronously, and then the position of the HWP can be accurately controlled so that the laser energy can be distributed as an apodization function along the axial direction of the fiber. By using the energy apodization method, the gratings with a reflectivity of 75% and a side lobe suppression ratio of 25 and 32 dB are fabricated in the fiber with a core diameter of 9 and 4.4 µm, respectively. The temperature and strain sensitivities of the energy-apodized fiber gratings with a core diameter of 4.4 µm are 10.36 pm/°C and 0.9 pm/µε, respectively. The high-reflectivity gratings fabricated by this energy apodization method are expected to be used in high-power narrow-linewidth lasers and wavelength division multiplexing (WDM) systems.

**Keywords:** half-wave plate; polarization beam splitter; apodized fiber gratings; energy apodization

## 1. Introduction

Fiber grating is a kind of diffraction grating formed by the axial periodic modulation of the refractive index of the fiber core through a certain method, and it is a passive filter device [1]. Fiber Bragg grating (FBG) is a typical reflective device with wavelength selectivity. As an all-fiber device, FBG has been widely studied and applied due to its advantages of low cost, fiber compatibility, and easy integration [2]. The uniform-period FBG, commonly called FBG, is one of the earliest and most widely used gratings. The modulation depth of the refractive index and the grating period are constant, and the wave vector direction of the grating is consistent with the axial direction of the fiber [3,4]. This kind of gratings has an important application value in fiber lasers, fiber sensors, a dense wavelength division multiplexing (DWDM) system, and a high-speed fiber sensing demodulation system [5–9]. There are many side lobes on both sides of the reflection spectrum of a uniform FBG with high reflectivity, which is caused by Fabry–Perot cavity resonance formed by an abrupt change of the refractive index at both ends of the grating. The existence of side lobes greatly reduces the wavelength selectivity of the FBG. At the same time, large side lobes in multiwavelength systems will cause crosstalk between adjacent channels, reduce the side lobe suppression ratio, and require large wavelength intervals in channels. If the time delay oscillation is too large, the dispersion compensation

is not good, which will restrict the application of the fiber grating. In order to suppress the side lobes in the reflection spectrum and reduce the time delay oscillation, the refractive index modulation amplitude can be changed. The apodization method is adopted to make the refractive index modulation amplitude conform to the distribution effect of the apodization function, which can effectively suppress the side lobes and improve the side lobe suppression ratio [10].

Using an apodized phase mask to make apodized fiber gratings is the most traditional method for fabricating apodized fiber gratings [11,12]. This method is relatively simple, but it is difficult to make phase masks, different apodization gratings need different phase masks, and the cost is high. At the same time, the average refractive index modulation coefficient of the fiber gratings produced in this way is not constant, and the short-wave oscillation effect will appear.

Femtosecond laser has the advantages of ultrashort pulse width and high peak power, which can be used as an excellent optical micro-nano processing method [13–15]. Based on the principle of multiphoton absorption, femtosecond laser can induce a permanent change of the refractive index in the transparent material [16,17], so the type II fiber gratings prepared by femtosecond laser have good temperature stability [18]. Femtosecond laser has important applications in the fabrication of high-temperature fiber gratings of quartz fiber and sapphire fiber [19,20]. The femtosecond laser point-by-point method has good flexibility and can be used to fabricate different types of gratings in different fibers [21–23]. Williams et al. proposed an oblique apodization method for preparing apodized FBGs, which was obtained by controlling the moving track of the fiber and changing the transverse position of the laser pulse modulation [24]. In this method, the refractive index modulation region is distributed obliquely in the core by using the point-by-point method, and the coupling efficiency at different positions of the core is adjusted to achieve the effect of apodization distribution of coupling strength.

In this paper, a new method for preparing apodized FBGs by femtosecond laser is proposed. In this method, the energy of the femtosecond laser pulse can be adjusted in real time by combining a half-wave plate (HWP) and polarization beam splitter (PBS). By controlling the rotation angle and speed of the stepping motor through the program, and then adjusting the position of the HWP, the laser pulse energy can be accurately controlled, and the refractive index modulation amplitude can conform to the apodization function distribution so as to achieve the effect of apodized FBGs. This method is flexible in design and not limited by core diameter and grating length, and can be used to design and fabricate apodized FBGs with different parameters.

## 2. Principle and Methods

The optical apodization of an FBG means that the amplitude of the photoinduced refractive index modulation in the grating is distributed along the axis of the fiber as a bell function. The coupling coefficient of this apodization envelope function is the largest at the center of the grating, and gradually decreases toward both ends, so that the refractive index variation amplitude at both ends of the FBG gradually tends to zero, which weakens the Fabry–Perot cavity resonance effect and reduces the side lobes of the FBG reflection spectrum. The FBG can be considered a periodic refractive index modulation region in the axial direction of the fiber under the action of periodic laser pulses. The photoinduced refractive index change $\Delta n$ inside the fiber core can be expressed by the following formula [25]:

$$\Delta n = \Delta n(z)\{1 + v\cos[\frac{2\pi}{\Lambda}z + \varphi(z)]\} \tag{1}$$

where $\Delta n(z)$ is the envelope function of refractive index change, $v$ is the fringe visibility of the index change, $\Lambda$ is the grating period, and $\varphi(z)$ describes the phase change of the grating.

Typical envelope functions include the Gaussian distribution function, the superGaussian distribution function, the rising cosine function, and the Cauchy distribution function. In this experiment, the Gaussian distribution function is taken as an example to make the core refractive index modulation amplitude conform to the Gaussian distribution function, and apodized FBGs are prepared. The Gaussian distribution function can be expressed by the following formula [25]:

$$\Delta n_{eff}(z) = \Delta n_{eff} exp(\frac{-4ln2z^2}{FWHM^2}) \tag{2}$$

where *FWHM* is full width at half maximum.

A device diagram of apodized fiber gratings prepared by the femtosecond laser point-by-point method based on energy regulation is shown in Figure 1. The femtosecond laser (light-conversion Pharos) emits 1030 nm laser with a pulse width of 290 fs and a repetition frequency of 200 kHz. The 1030 nm femtosecond laser is frequency-doubled to a 515 nm femtosecond laser by a nonlinear crystal (β-BaB$_2$O$_4$) (BBO). The 515 nm femtosecond laser is focused through a 60× high numerical-aperture oil-immersion objective lens (Olympus, NA = 1.42) on the center of the fiber core placed on the five-dimensional processing platform (Aerotech).

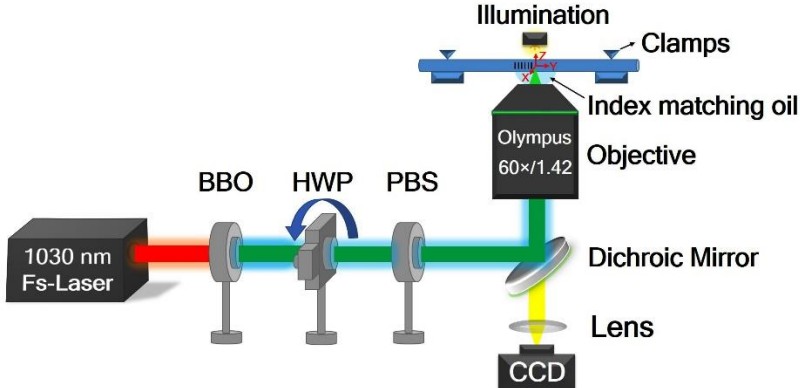

**Figure 1.** Schematic diagram of the device for fabricating apodized FBGs based on energy regulation method.

In this way, the laser pulses can be focused on a smaller focal point, which can improve the processing accuracy. The laser energy regulation device is included in the optical path. The HWP is fixed on the rotating bracket of the stepping motor controlled by the program, and the laser polarization can be changed in real time. The laser pulse energy can be precisely controlled by the combination of the PBS with a high extinction ratio. The laser pulse energy corresponding to different angles of the HWP can be accurately calibrated by a laser power meter (OPHIR 10A-P). By inputting an apodization function model through the program, the laser pulse energy can be distributed as a Gaussian function along the fiber axis. The rotating speed of the stepping motor can be synchronized with the moving speed of the processing platform so that the refractive index modulation amplitude will be distributed as an apodization function in the middle of the fiber core, and apodized FBGs with different parameters can be flexibly prepared.

## 3. Design Results and Discussion

### 3.1. Spectral Comparison

The FBG phase matching condition is as follows [26]:

$$m\lambda_B = 2n_{eff}\Lambda \tag{3}$$

where $\Lambda$ is the grating period, $n_{eff}$ is the effective refractive index, $\lambda_B$ is the wavelength, and $m$ is the grating order. A second-order apodized FBG is fabricated in a 9 μm core diameter fiber with a grating period of 1.071 μm and a central wavelength of 1550 nm. An apodized FBG is fabricated in a 4.4 μm core diameter fiber with a grating period of 1.064 μm and a center wavelength of 1030 nm. Because the center wavelength is short, the grating is a third-order apodized FBG.

The micrographs of apodized FBGs prepared by oblique apodization and energy apodization methods are shown in Figure 2. Figure 2a,b shows micrographs of apodized fiber gratings prepared by different methods in the fiber with a core diameter of 9 μm. Figure 2c,d shows micrographs of apodized fiber gratings prepared by different methods in the fiber with a core diameter of 4.4 μm. In order to make a clear comparison between the two methods, the grating length is selected as 50 μm. It can be seen from Figure 2 that femtosecond laser pulses with ultrahigh peak power are focused on the fiber core by the point-by-point method to form a periodic refractive index modulation region. The laser spot size is about 500 nm. Figure 2a,c presents apodized FBGs prepared by an oblique apodization method, which keeps the laser energy unchanged and changes the transverse position of the laser pulse modulation by controlling the moving track of the optical fiber. Figure 2b,d shows apodized FBGs prepared by an energy regulation method. In this method, the laser pulse energy needs to be adjusted in real time so that the refractive index modulation amplitude is distributed as an apodization function along the fiber axis.

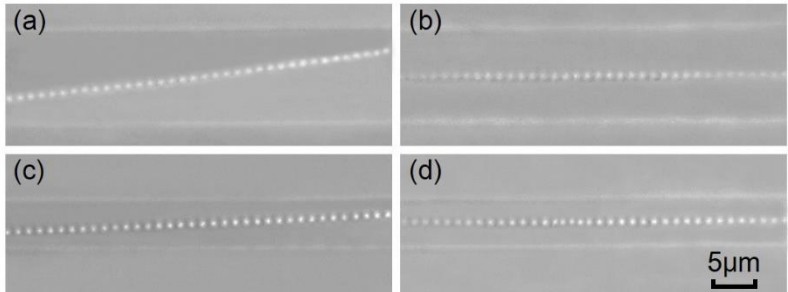

**Figure 2.** (**a**,**b**) shows micrographs of apodized fiber gratings fabricated by different methods in a fiber with a core diameter of 9 μm. (**c**,**d**) shows micrographs of apodized fiber gratings fabricated by different methods in a fiber with a core diameter of 4.4 μm.

When preparing FBGs with the same reflectivity, the maximum single-pulse energy required for preparing apodized FBGs is larger than that for uniform FBGs. This is because the refractive index modulation amplitude of the apodized grating decreases gradually along both ends of the grating, which leads to the decrease of the average coupling efficiency of the grating. Therefore, in order to obtain FBGs with the same reflectivity, the maximum single-pulse energy needed to prepare apodized FBGs will be higher. The single-pulse energies needed for preparing uniform FBGs and apodized FBGs with a 75% reflectivity are 90 and 100 nJ (maximum single-pulse energy), respectively. The length of all gratings prepared is 4 mm. The moving speed of the processing platform is 0.4284 mm/s, and the time required is about 10 s.

Three different types of FBGs are prepared in a single-mode fiber (Corning, SMF-28e), and a spectral comparison is shown in Figure 3a. The side lobe suppression ratios of the three FBGs are 10, 16, and 25 dB. The apodized grating with a higher side lobe suppression ratio is fabricated by an energy regulation method. The reflectivity and FWHM of the apodized grating are 75% and 0.31 nm, respectively. Figure 3b shows the preparation of apodized FBG with higher reflectivity (>99%). The side lobe suppression ratios (SLSR) before and after apodization are 8 and 18 dB, respectively. It can be seen that with the increase of reflectivity, the effect of side lobe suppression will be reduced.

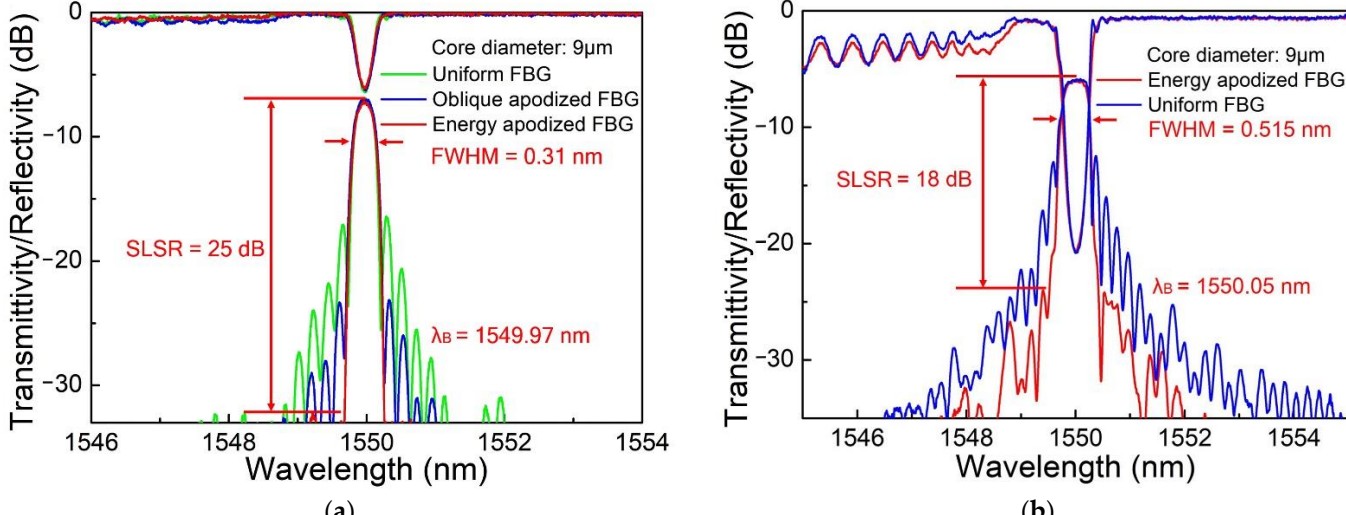

**Figure 3.** (**a**) Comparison of a uniform FBG spectrum and an apodized FBG spectrum with a reflectivity of 75% fabricated by two methods in fibers with a core diameter of 9 μm. (**b**) Comparison of a uniform FBG spectrum and an apodized FBG spectrum with a reflectivity of 99% fabricated by an energy regulation method in fibers with a core diameter of 9 μm.

Compared with a standard single-mode fiber, a thin core fiber has a smaller cut-off wavelength and can support shorter-wavelength light to realize single-mode transmission, so it has a good application value in short-wavelength laser. However, the core diameter of a thin core fiber is small, so it is more challenging to prepare apodized FBGs with different parameters. FBGs have important applications in fiber lasers [27,28], and the wavelength of 1030 nm is a common laser wavelength. As shown in Figure 4a, three different types of FBGs are prepared in a 4.4 μm core diameter fiber (Nufern 780-HP) by an energy regulation method. The reflectivity of the three FBGs is 75%, and the side lobe suppression ratios are 10, 20, and 32 dB, respectively. The FWHM of the FBG prepared by the energy regulation method is 0.18 nm. Figure 3b shows the preparation of an apodized FBG with a reflectivity of over 99%. The side lobe suppression ratios before and after apodization are 7 and 22 dB, respectively.

Due to the smaller core diameter of a thin core fiber, the energy density of the optical field in the core is higher, and the coupling efficiency will be higher. Compared with the standard single-mode fiber, when the grating with the same reflectivity is prepared, the laser pulse energy needed to prepare the grating with the same reflectivity will be smaller. In the fiber with a core diameter of 4.4 μm, the single pulse energies required to prepare uniform FBGs and apodized FBGs with a reflectivity of 75% are 60 and 70 nJ (maximum single pulse energy), respectively. The length of all gratings prepared is 4 mm. The moving speed of the processing platform is 0.4256 mm/s, and the time required is about 10 s.

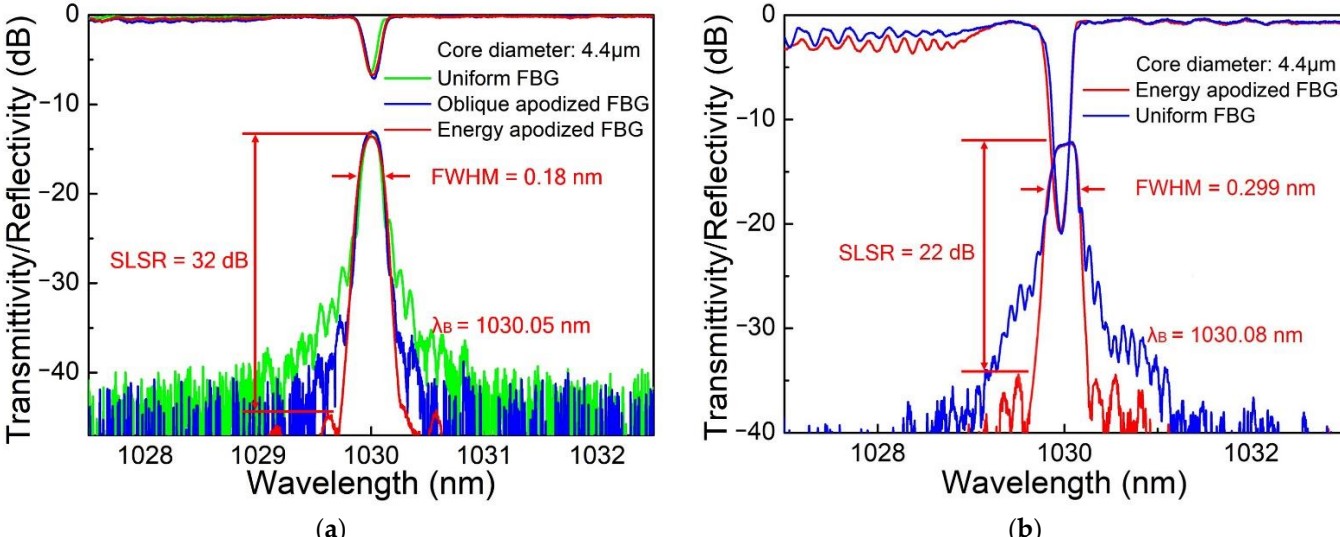

**Figure 4.** (**a**) Comparison of a uniform FBG spectrum and an apodized FBG spectrum with a reflectivity of 75% fabricated by two methods in fibers with a core diameter of 4.4 μm. (**b**) Comparison of a uniform FBG spectrum and an apodized FBG spectrum with a reflectivity of 99% fabricated by an energy regulation method in fibers with a core diameter of 4.4 μm.

### 3.2. Apodized FBG Temperature Testing

The temperature and strain characteristics of apodized FBGs prepared in an optical fiber with a core diameter of 4.4 μm are tested by combining a spectrometer (Yokogawa, AQ6370D), broadband light source (NKT Photonics), and customized optical fiber coupler. First, the grating is annealed in a tube furnace at 800 °C for 3 h, then cooled to room temperature, in order to eliminate the internal stress and unstable structure after femtosecond laser processing. Then, the apodized grating is tested again from room temperature to 800 °C with an interval of 100 °C. Each temperature point is kept for 30 min, and the corresponding spectral data are recorded. The drift curve of the resonance wavelength with temperature is shown in Figure 5. With the increase of temperature, it shows red shift characteristics, which is consistent with the characteristics of quartz FBGs.

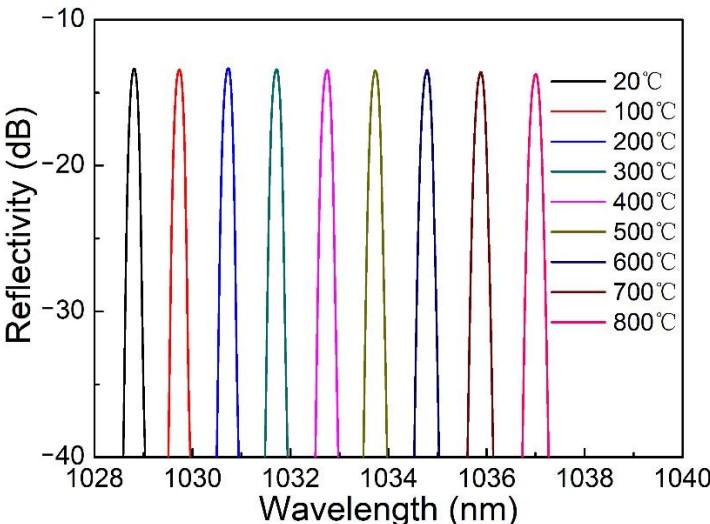

**Figure 5.** The reflection spectrum of an apodized FBG with a core diameter of 4.4 μm varies with temperature.

The measured data are fitted linearly, and the fitting curve is shown in Figure 6. The temperature sensitivity of the apodized FBG is 10.36 pm/°C. The experimental results show that the apodized FBG has good temperature stability and can work stably at 800 °C.

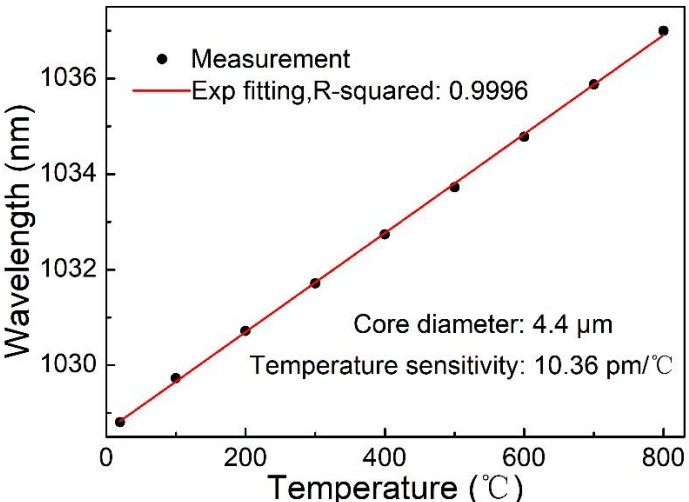

**Figure 6.** The relationship between the resonant wavelength of an apodized FBG with a core diameter of 4.4 μm and temperature.

### 3.3. Apodized FBG Strain Testing

The apodized FBG is placed on a stress-testing device (WDW-100) to prepare for strain testing. The apodized FBG is clamped on the testing machine by the fiber clamps, and the stress applied by the testing machine is controlled by the program. The maximum test stress is 1 N, and the stress test interval is 0.1 N. When the spectrum is stable, record the corresponding spectral data. The drift curve of the resonance wavelength with stress is shown in Figure 7.

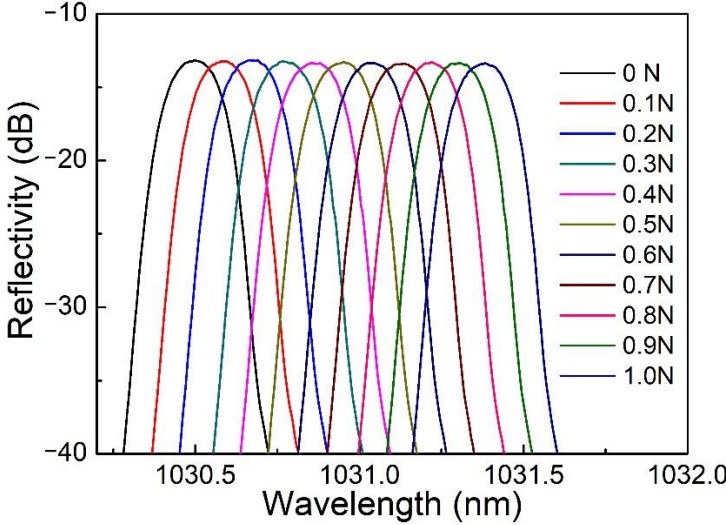

**Figure 7.** The reflection spectrum of an apodized FBG with a core diameter of 4.4 μm varies with strain.

With the increase of stress, the spectrum is red-shifted. Axial strain $\varepsilon$ can be obtained by the following formula:

$$\varepsilon = F/\pi r^2 E \tag{4}$$

where $F$ is the axial stress, $r$ is the fiber radius ($r$ = 62.5 μm), and $E$ is Young's modulus ($E$ = 73 GPa). The linear relationship between the resonance wavelength and strain of an apodized FBG is fitted in Figure 8, and the strain sensitivity is 0.9 pm/με.

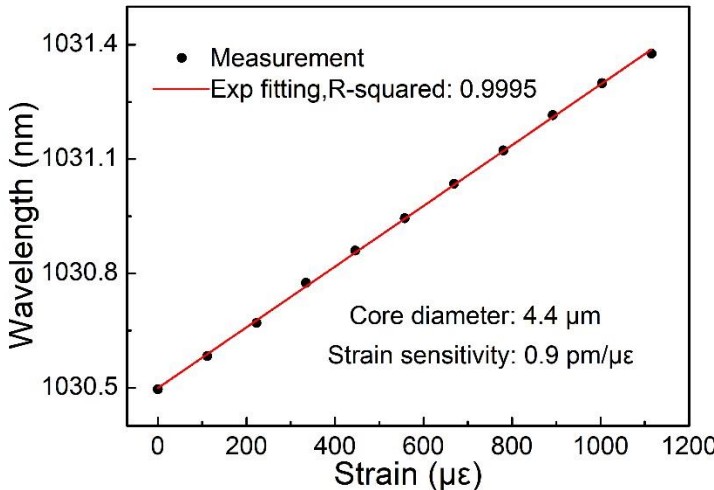

**Figure 8.** The relationship between the resonant wavelength of an apodized FBG with a core diameter of 4.4 μm and strain.

## 4. Conclusions

In conclusion, a flexible and efficient method for fabricating apodized FBGs is proposed, which is to control energy through the combination of an HWP and PBS. The stepping motor is controlled by a program, and the position of the HWP is rotated synchronously to adjust the energy size. In different core diameter fibers, the point-by-point method is used to make the core refractive index modulation distributed as an apodization function. The spectral effects of apodized fiber gratings fabricated by oblique apodization and energy apodization methods are compared. When the energy apodization method is used to fabricate apodized fiber gratings, it does not need to change the processing trajectory obliquely, so the design is more flexible and better spectral quality can be obtained. In particular, apodized FBGs with a 75% reflectivity and a 32 dB side lobe suppression ratio are fabricated in a 4.4 μm core diameter fiber. The temperature sensitivity and strain sensitivity are 10.36 pm/°C and 0.9 pm/με, respectively. The apodized fiber gratings fabricated by this method have potential applications in high-power narrow-linewidth lasers and wavelength division multiplexing (WDM) systems.

**Author Contributions:** Conceptualization, Q.G. and Z.Z.; methodology, Q.G.; software, Q.G. and X.P.; formal analysis, C.C. and Z.T.; data curation, Q.G., S.L. and B.W.; writing—original draft preparation, Q.G.; writing—review and editing, Y.Y. All authors have read and agreed to the published version of the manuscript.

**Funding:** National Natural Science Foundation of China (91860140, 62090064, 61874119); Science and Technology Development Project of Jilin Province (20180201014GX).

**Institutional Review Board Statement:** Not applicable.

**Informed Consent Statement:** Not applicable.

**Data Availability Statement:** The data presented in this study are available on request from the corresponding author.

**Acknowledgments:** This work is supported by the National Natural Science Foundation of China (91860140, 62090064, 61874119) and the Science and Technology Development Project of Jilin Province (20180201014GX).

**Conflicts of Interest:** The authors declare no conflict of interest.

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
