# Peer review of "Femtosecond Laser Fabricated Apodized Fiber Bragg Gratings Based on Energy Regulation"

_photonics, doi:10.3390/photonics8040110_

Round 1

Reviewer 1 Report

Based on a fs laser and the point by point inscription technique, a fabrication method for apodized FBGs is reported by varying the pulse energy along the length of the FBG. When compared with the earlier reported oblique method, improved side lobe suppression (25dB, 32dB) is demonstrated for FBGs in two different types of optical fibers. To my knowledge, this approach shows novelty and important advantages compared to approaches used earlier, not only with regard to side lobe suppression but also to ease of fabrication. Therefore, I recommend the manuscript to be published but I recommend some minor changes:
Figures 3, 4 and 7: The title of the ordinate is not correct and confusing. Intensity can’t go with “dB”, for “Transmission” or “Reflectivity” dB would be appropriate. Are the reflected spectra given as reflected power level in dBm?
Page 4, lines 156 ff: The author mention that they increased the pulse energy from 90nJ to 100nJ to achieve the same reflectivity of 75% for uniform and apodized gratings. The authors should also mention the length of the three types of gratings.
The experimental setup and procedure for measuring the strain response should be described in more detail. The values used in Eq 4 (for the Young’s module, and fiber diameter) should be given.
Abstract: I recommend to use the term side lobe instead of side mode.

Reviewer 2 Report

For this manuscript, my comments are as below:

  1. Authors should describe the laser spot size for Figure 2 and describe the spot size effect in grating characteristics for the two different core diameters.
  2. For the FBG applications in optical communications, the more than 99% reflectivity is generally needed for applying in the Add-Drop filter. Authors should show the side mode rejection for the strong grating (more than 99 % reflectivity).
  3. Does it take how much time for fabricating a 1-cm length grating by using the point-by point technique? Please authors describe it.

Therefore, I think this manuscript requires the minor revision for publishing in Photonics.

Reviewer 3 Report

15:
“…suppression ratio of high reflectivity fiber Bragg gratings (FBGs) fabricated by femtosecond laser.”
Actually, in the manuscript the authors deal only with mid reflective FBGs (R ~ 75%). I think that experimental demonstration of high reflective FBGs (R > 95%) should be carried out in order to determine the performance capabilities of the inscription method. This is important because for high reflective gratings the degree of suppression of side lobes will decrease regardless of the apodization method and function.
16:
“Compared with the oblique apodization method, the energy apodization method is more flexible. By inputting different types of apodization functions through the program, the rotation speed of the stepping motor can be adjusted synchronously, and then the position of the HWP can be accurately controlled, so that the laser energy can be distributed as apodization function along the axial direction of the fiber.”
I think that the authors gloss over the real things. In particular, the rotation of the HWP modulates the energy of fs laser pulses with a harmonic function. Further, the energy of laser pulses is related to the modulation of the refractive index in a nonlinear manner. Accordingly, the calculation of the transfer function is required to accurately define the apodization profile. The situation is similar with oblique apodization method, where the transfer function should be calculated in order to relate the modulation of the refractive index to the position of the focal point relative to the core.
50:
“Apodized FBGs have been widely used in DWDM system and high-speed fiber sensing demodulation system [8-10].”
This appendix duplicates an earlier thought (lines 37-38). References 9 and 10 are identical.
52-61:
I think that this part can be significantly shortened as far as the focus of the manuscript lays on the femtosecond-pulse based inscription method.
74:
“However, this method is limited by the core diameter and grating length, which can only adjust the transverse position in a small range, and cannot customize the apodization function effect.”
Actually, the authors invent these limitations. The transverse position of the fiber core can be changed in quite large ranges (> 100 μm) with very small steps (~ 10-100 nm) by using piezo or air-bearing stages. The apodization function can also be customized, as it was shown in [R. J Williams, R. G. Krämer, S. Nolte, M. J. Withford, and M. J. Steel, “Detuning in apodized point-by-point fiber Bragg gratings: insights into the grating morphology,” Opt. Express, vol. 21, no. 22, p. 26854, Nov. 2013, doi: 10.1364/OE.21.026854.].
101:
“…Caussian function”
Caussian -> Gaussian
136:
“A second-order apodized FBG is fabricated in a 9μm core diameter fiber with grating period of 1.071μm and central wavelength of 1550nm. Apodized FBG is fabricated in a 4.4 μm core diameter fiber with grating period of 1.064 μm and center wavelength of 1030 nm.”
It is not clear why the authors deal with 2-nd and 3-rd order FBGs, because they use high-NA objective and second harmonic of the fs laser, allowing them to inscribe 1-st order gratings. Will the order of resonance affect the efficiency of apodization?
170:
“The reflectivity and FWHM of the apodized grating are 75%...”
Probably the authors lost second vertical axis of the graph. In particular, reflection coefficient of 75% corresponds to -1.25 dB, but on the graph reflection reaches only -7 dB. This problem is also actual for the following spectra. Instead of “intensity” word, it would be correct to use “reflection” and “transmission” words.
Other:
-    The length of the inscribed FBGs should be indicated.
-    The authors extensively cite papers of their Chinese colleagues and some of these citations do not adequately represent the World experience (for example, the first block of citations [1-4], as well as [23], [24], [25], and [26], where the authors cite random papers which do not have a very faint relation to the topic).

Round 2

Reviewer 3 Report

I appreciate the efforts of the authors to improve the manuscript; however, I still believe that the authors make a mistake in normalizing the reflection spectra. As an example, I am attaching a figure with the reflection / transmission spectra of a uniform grating, which was obtained using a fairly common software package (Optiwave OptiGrating). In this figure, you can see how the spectra should look with correct normalization. Another remark is related to the use of the SNR (=signal-to-noise ratio) abbreviation in the figures. I propose to replace it with SLSR (=side lobe supression ratio), since this term is used in the text of the manuscript.
